# Capacitive Electrode-Based Electric Field Treatments on Redox-Toxic Iron Deposits in Transgenic AD Mouse Models: The Electroceutical Targeting of Alzheimer’s Disease Feasibility Study

**DOI:** 10.3390/ijms24119552

**Published:** 2023-05-31

**Authors:** Younshick Choi, Won-Seok Lee, Jaemeun Lee, Sun-Hyun Park, Sunwoung Kim, Ki-Hong Kim, Sua Park, Eun Ho Kim, Jong-Ki Kim

**Affiliations:** 1Departments of Biomedical Engineering & Radiology, School of Medicine, Daegu Catholic University, Daegu 42472, Republic of Korea; younshick@cu.ac.kr (Y.C.); jahwasu3@cu.ac.kr (S.K.); 2Departments of Biochemistry, School of Medicine, Daegu Catholic University, Daegu 42472, Republic of Korea; leeee@cu.ac.kr; 3Korea R&D Center for Advanced Pharmaceuticals & Evaluation, Korea Institute of Toxicology, Daejeon 34114, Republic of Korea; jaemeun.lee@kitox.re.kr (J.L.); sunhyun.park@kitox.re.kr (S.-H.P.); 4Department of Optometry and Vision Science, Daegu Catholic University, Kyungsan 38430, Republic of Korea; kkh2337@cu.ac.kr; 5Department of Neurology, Inje University Ilsan Paik Hospital, Koyang 10380, Republic of Korea; sua0404@naver.com

**Keywords:** electroceutical, intraplaque magnetite, alternating electric field, electro-Fenton effect, Alzheimer’s disease

## Abstract

Iron accumulation in the brain accelerates Alzheimer’s disease progression. To cure iron toxicity, we assessed the therapeutic effects of noncontact transcranial electric field stimulation to the brain on toxic iron deposits in either the Aβ fibril structure or the Aβ plaque in a mouse model of Alzheimer’s disease (AD) as a pilot study. A capacitive electrode-based alternating electric field (AEF) was applied to a suspension of magnetite (Fe_3_O_4_) to measure field-sensitized reactive oxygen species (ROS) generation. The increase in ROS generation compared to the untreated control was both exposure-time and AEF-frequency dependent. The frequency-specific exposure of AEF to 0.7–1.4 V/cm on a magnetite-bound Aβ-fibril or a transgenic Alzheimer’s disease (AD) mouse model revealed the degradation of the Aβ fibril or the removal of the Aβ-plaque burden and ferrous magnetite compared to the untreated control. The results of the behavioral tests show an improvement in impaired cognitive function following AEF treatment on the AD mouse model. Tissue clearing and 3D-imaging analysis revealed no induced damage to the neuronal structures of normal brain tissue following AEF treatment. In conclusion, our results suggest that the effective degradation of magnetite-bound amyloid fibrils or plaques in the AD brain by the electro-Fenton effect from electric field-sensitized magnetite offers a potential electroceutical treatment option for AD.

## 1. Introduction

The neurons located in the locus coeruleus (LC) are particularly prone to generating Aβ as a protective response [1] and have been found to be vulnerable to amyloid accumulation and tauopathy. This accumulation can occur up to several decades prior to clinical manifestation. As the Braak stage increases, the tau pathology appears to spread to the forebrain, including the entorhinal cortex [2,3] (stage 3–4), eventually resulting in the accumulation of toxic tauopathies in the nucleus basalis of Meynert (NBM), which leads to clinically apparent cholinergic failure [4] (stage 5–6).

Early studies have found that iron overload is directly proportional to cognitive decline associated with Alzheimer’s disease (AD) [5]. Iron deposits were found not only in and around senile plaques [6] but also in the sites of cortical tau accumulation [7]. Iron deposition can regulate tau phosphorylation by inducing the activity of multiple kinases that promote tau hyperphosphorylation and the aggregation of hyperphosphorylated tau, possibly through direct interaction via a putative iron-binding motif in the tau protein, facilitating the formation of neurofibrillary tangles (NFT) [8]. Furthermore, in human studies, high levels of iron have been reported to colocalize with tau in NFT-bearing neurons [7], and ferrous magnetite was identified as an iron species present in Aβ-plaque [6,9]. Other redox metal ions such as Zn^2+^ and Cu^2+^ were bound to Aβ plaques, indicating their role as redox centers [6,10] and peripheral halo of neurotoxic soluble Aβ-oligomer surrounding plaque [11,12]. Recent studies in the tau cell model showed that insoluble tau aggregates exhibited neurotoxic effects on neuronal cells [13,14]. An iron deposit was integrated into protein aggregates by binding to peptide, generating redox-toxic ferrous iron [6,15]. Therefore, iron overload drives a series of events, including microglial activation by uptake of iron, reactive astrocytes induction by ROS generation, promotion of Aβ plaque and tau tangles by iron binding, and ferroptosis-mediated neuronal or astrocyte loss, which further aggravates the disease [16,17,18,19]. We hypothesize that iron deposits in both insoluble Aβ plaques and tau tangles, if not eliminated, are continuously neurotoxic. Therefore, irrespective at what stage AD is diagnosed, treatment should include either blocking amyloid and tau spreads or curing pre-existing iron deposits with protein aggregates to prevent further neuronal damage [16,17,20,21]. Direct glial uptake of iron or ROS released from ferrous iron deposits activates microglial cells and astrocytes [16,17,22], inducing reactive astrocytes [22] or ferroptosis of astrocytes depending on the amount of iron-mediated toxicity via the NOX4 pathway [23].

Potential therapeutic approaches for minimizing iron toxicity include the use of N-acetyl cysteine, nonsteroidal anti-inflammatory compounds, and iron chelation [16]. Our previous proof-of-concept study demonstrated that the electron-release activity of proton-stimulated magnetite prevents cognitive decline, suggesting that targeting intraplaque magnetite might be a feasible treatment option for AD [21]. Proton-sensitized electron emission made breaking up covalent bonding in both Fe_3_O_4_-Aβ and protein matrix disrupting insoluble Aβ plaque that led to the removal of toxic iron deposits and protein aggregates and induced cognitive improvements. Here, we introduce another therapeutic approach that utilizes the electric field-sensitized electro-Fenton effect from protein-bound magnetite to eliminate iron toxicity by breaking the bond between ferrous iron and proteins.

The conventional electric contact method uses a current source and electrodes that are in contact with the head to produce an electric field within the brain [24]. Modulating iron deposition in the LC or NBM may help control degeneration in AD-susceptible anatomical areas and block the spread of toxic tauopathies. We designed a capacitive electric field method that uses a source of voltage and electrodes without direct injection of current into the head to easily deliver an electric field in a transcranial manner even within the deep brain. Magnetite is a semiconductor (band gap ~0.2 eV) at low temperatures; however, at room temperature, magnetite exhibits properties similar to those of a metal (250 Ω^−1^ cm^−1^) [25]. Thus, magnetite exhibits electro-Fenton effects driven by a static electric field, and the resultant common reactive oxidizing species (ROS) (e.g., •OH) induces unselective oxidation of various toxic organic pollutants. Therefore, this process is routinely used for wastewater remediation [26,27]. Moreover, the generation of free radicals could be accelerated in the magnetite solution under alternating magnetic field induction [28]. This study aimed to investigate the electro-Fenton effects sensitized by an alternating electric field (AEF) on magnetite in an aqueous solution, a magnetite-bound amyloid fibril, and in vivo protein magnetite deposition as a pilot transcranial treatment in a transgenic AD mouse model.

## 2. Results

### 2.1. ROS Measurement

Since magnetite is known to be a dielectric dispersive material, ROS yield may depend on the frequency of the electric field. Hence, we first investigated the appropriate frequency of AEF that should be applied to magnetite-bound fibril or Aβ plaque in the AD mouse model. A relatively higher level of ^•^OH production was observed at 1 MHz AEF than at 220 or 1.5 kHz AEF in the presence of Fe_3_O_4_ nanoparticles using APF oxidant fluorescence (Figure 1). The enhancement of superoxide, observed using DHE oxidant fluorescence, was relatively minimal compared to that of ^•^OH.

### 2.2. AEF Treatment on Aβ-Fe_3_O_4_ or Aβ Alone Fibril

The integrity of both fibrils was monitored via thioflavin-T fluorescence (Tf-T) using sampled aliquots of the reaction solutions at various times (Figure 2). Tf-T fluorescence steadily decreased in the Aβ-Fe_3_O_4_ fibril but did not significantly change in the Aβ fibril over 60 min during AEF exposure.

The TEM image of both fibril samples treated for 30 min presented degradation with fragmentation occurring in the Aβ-Fe_3_O_4_ fibril, compared to the untreated control; however, the morphology of the Aβ-only fibril remained almost entirely intact (Figure 3), suggesting an optimal time period for AEF treatment in the following in vivo experiment, which may cause almost no damage to the protein matrix of the surrounding brain tissue.

Since magnetite was bound to the Aβ peptide, networked fibrils were relatively enhanced, as seen in Figure 3A. A TEM image obtained after AEF treatment showed either disruption of the covalent bonding between Aβ and magnetite or fragmentation of the Aβ-peptide strand. The patterns of fibril fragmentation appeared similar to those of the Aβ-magnetite fibril treated with proton stimulation, where electron emission was the dominant process responsible for breaking the chemical bonding in the magnetite-bound fibril [15,21]. This irreversible change in insoluble amyloid fibrils yields biological consequences, such as the conversion of toxic ferrous magnetite into nontoxic ferric magnetite and soluble fibril fragments, which may facilitate the activation of microglial phagocytosis.

### 2.3. AEF Treatment on AD Mouse Model

To obtain the effective dose value of the electric field, AEF treatment with either HD-F (1.4 V/cm, 30 min, n = 3) or LD-F (0.7 V/cm, 30 min, n = 3) was performed on 5xFAD mice, and the results of the histological analysis were compared to the untreated AD control (n = 3). In each mice group, quantitative measurements were made by counting either all the intraplaque magnetite or the Aβ-plaque burden in adjacent tissue sections (n = 3) containing the entorhinal cortex–hippocampal plane. Turnbull staining revealed the presence of typical intraplaque ferrous magnetite in the brain of the untreated AD control mice (Figure 4A), which was notably reduced by 60–80% in a dose-dependent manner in AEF-treated mice (Figure 4B–D). The Aβ-plaque burden post-EFT was reduced at a similar rate in the Congo red staining analysis.

To examine the effects of AEF treatment on Aβ-burden reduction and memory improvement, RAM and NOR behavioral tasks were performed on the following mice groups: untreated WT-cont (n = 3), WT-post-EFT (HD, n = 3), Tg-pre-EFT (n = 8) which included five 5xFAD and three 3xTG; and Tg-post-EFT (n = 8) which included five 5xFAD (HD) and three 3xTG (HD). Animals were tested using memory behavioral tasks and post mortem measurements of pathological markers were obtained. AEF treatment was performed three weeks prior to the post-EFT behavioral test. This period of 3 weeks was chosen based on the results of our prior work [21] that showed at least 3–4 weeks were required to obtain improved cognitive function after proton-stimulation treatment targeting intraplaque magnetite.

During the NOR task, all AD mice prior to AEF treatment (pre-EFT) exhibited a decrease in the time values for the novel object recognition test compared to WT normal mice (Figure 5A). On average, AEF-treated mice (post-EFT) spent relatively longer periods of time on the novel object than the pre-EFT mice group (*p* < 0.001), yielding a high discrepancy index (Figure 5). This trend presented a minor difference between the 5xFAD and 3xTG AD mice. The time spent by AEF-treated WT in the novel object recognition did not differ significantly from that spent by untreated WT-cont. During the RAM test, all AD mice prior to AEF treatment (pre-EFT; 5xFAD and 3xTG) exhibited relatively higher error rates than the WT normal mice; however, the frequency of error rates significantly decreased in the post-EFT AD mice than in the pre-EFT AD mice (*p* < 0.01) (Figure 6). AEF-treated WT mice did not show significant differences in error rates compared to the untreated WT-cont.

### 2.4. Assessment of Overall Neuronal Morphology in the AEF-Treated Brain

Thy1-YFP-H transgenic mice express a yellow fluorescent protein (YFP) at high levels in motor and sensory neurons, as well as subsets of central neurons. To test whether our setup was safe for use in a normal mouse brain when exposed to AEF treatment, we used a tissue-clearing-based histological assessment of both untreated and AEF-treated WT mouse brains. We performed transcardiac perfusion and 4% PFA fixation on the mice and cleared 2 mm of sliced mouse brain within 24 h of treatment.

Subsequently, YFP fluorescence in the cleared 2 mm thick mouse-brain slices was imaged and 2D images were converted to 3D for assessing the changes that occur in the neuronal structure (Figure 7).

Fluorescent photomicrographs revealed YFP-positive neurons in the cortex, hippocampus, and amygdala regions (anterior coronal view, Figure 7A; posterior coronal view, Figure 7B). High magnification image revealed the presence of fine neuronal structures, including the dendritic spines, in the pyramidal neurons of the cortex (Figure 7C,D), hippocampus (Figure 7E), and amygdala (Figure 7E). These results demonstrate that no significant damage occurred to the tissue morphology or neuronal architecture of the AEF-treated mouse brain.

## 3. Discussion

In this study we found that a frequency-specific alternating electric field was able to sensitize electron emissions from the magnetite surface, inducing the electro-Fenton effect in an aqueous solution, to disrupt insoluble amyloid plaque containing redox-toxic magnetite. The exact origin of the magnetite integration into the Aβ plaque remains unknown. A few studies have suggested that these are generated by the interaction between Aβ and ferritin, which contains iron oxide nanoparticles [29,30]. However, not all Aβ-plaques contain magnetite. The results from our previous study [21] estimated that >70% of Aβ plaques contain magnetite, which is concurrent with other studies [31]. AEF-sensitized magnetite produced high levels of hydroxyl radicals, particularly at a higher frequency of the applied electric field. This effect might be due to the electro-Fenton effect on the magnetite solution caused by the static electric field [26], suggesting that the potential induction of the bipolar electrochemistry [26] is transiently set up by the rapidly changing field and dielectric dispersion property of magnetite [32]. In turn, the enhancement of the electro-Fenton effect can be ascribed to the increased electron transport activity and AC conductivity of magnetite with the increasing AEF frequency [33,34]. AEF induces electron release from the magnetite surface and the formation of ROS near the magnetite surface within the context of insoluble Aβ plaque. Therefore magnetite-derived electrons and hydroxyl radicals are prone to be consumed mainly in breaking Aβ plaque-magnetite bonds (Figure 3 and Figure 4) because the Aβ plaque matrix is covalently bonded with the magnetite.

Since AEF can be delivered to the deep brain in a noncontact transcranial manner with capacitive electrodes, targeting magnetite particles in protein aggregates using AEF therapy might serve as an electroceutical treatment option for AD patients. Hence, testing the AEF-driven electro-Fenton effect on the magnetite fibril in vitro or within the AD deep brain in vivo is crucial. The differential time values for inducing a disruption between Aβ-magnetite and Aβ-alone fibrils, as shown in the Tf-T fluorescence data presented in Figure 3, were used to determine 30 min as the optimum exposure time for AEF treatment while showing the electro-Fenton effects of magnetite on the fibril. The fragmentation and disruption of the fibrils, as shown in the TEM images in Figure 4, suggests a direct effect of AEF-driven electron emissions from the magnetite surface on the covalent bonding of Aβ proteins, as observed in our previous studies [15,21]. Thus, the generation of both electron and hydroxyl radicals from the AEF-exposed magnetite may not only break the bond of the Aβ–magnetite complex but also the Aβ protein matrix in the fibril. Based on these results, we administered AEF to a transgenic AD mouse model at different doses: 0.7 and 1.4 V/cm for 30 min. The dose-dependent degradation and reduction in both intraplaque iron species and Aβ plaque are shown in Figure 5. The entorhinal–hippocampal section demonstrated the effects caused by the interaction of AEF with intraplaque magnetite in the AEF-treated mouse model. This result indicates the potential therapeutic effects of transcranial AEF administration in the deep brain, potentially such as the locus coeruleus–norepinephrine system, which is an important target site for direct electroceutical stimulation in early onset AD [20]. This result is consistent with the results of proton-stimulation (PS) therapy that targeted intraplaque magnetite, as described in our previous study [15,21]. These studies together suggest that similar biological end-effects are elicited by both brain-stimulation tools targeting the common intraplaque magnetite. Moreover, the AEF-driven removal of toxic iron deposits suggests the potential beneficial effects of AEF treatment in preventing either monoamine oxidase B activity or reactive astrocytes formation, which has been implicated as a major culprit to cause AD [35]. Ongoing PS study revealed that a cascade of disease-modifying events, such as downregulating neuroinflammation, reactive astrocyte, and blocking tau spreading that is helpful to cognitive improvement, were triggered following PS-derived removal of a toxic iron deposit together with amyloid plaque (Data were prepared for publication in another journal). AD mice demonstrated clearly impaired memory function compared to WT control mice (Figure 6 and Figure 7) and both AD-mouse species demonstrated an improvement in cognitive function following AEF treatment (*p* < 0.001). Delivery of different types of physical energy on magnetite involved common electron emissions from the surface of iron oxide nanoparticles, which induced the fragmentation of insoluble Aβ plaque containing ferrous magnetite. The degradation of Aβ plaque into smaller-sized peptides and isolated magnetite increases the solubility of plaque and facilitated microglial phagocytosis-mediated clearance.

WT-normal mice did not show significant differences in behavioral tasks before and after AEF treatment, suggesting that the therapy did not induce any detrimental neuronal damage, which was consistent with the results of the tissue-clearing-based 3D-imaging analysis of AEF-treated brain tissue.

## 4. Materials and Methods

### 4.1. Synthesis of Magnetite and Aβ/Magnetite or Aβ Fibril

Magnetite was synthesized using a standard coprecipitation method. Aβ (1–42) peptide was obtained from a commercial manufacturer (AnyGen, AGP-8338, Seoul, Republic of Korea). Monomeric synthetic Aβ (1–42) was dissolved in 0.1 M NaOH to produce a 220 µM Aβ-stock solution and was left to stand for 30 min to ensure complete peptide dissolution. The Aβ solution was mixed with 80 µg/mL magnetite solutions to generate Aβ/magnetite solution. Aβ solutions with and without magnetite were neutralized to pH 7 with 0.5 M hydrochloric acid. The final Aβ and metal concentrations were 110 µM and 40 µg/mL, respectively. The Aβ and Aβ/magnetite solutions were incubated for 144 h at 40 °C.

### 4.2. Electric Field Stimulation

An AEF was generated using a functional generator and voltage amplifier. A capacitive electrode, as shown in Figure 8B, was manufactured on a flexible printed circuit board in which a pair of semispiral wires were sandwiched between polyimide films and connected to an electric field source to stimulate magnetite in a phantom or AD-mouse brain. AEF frequency values were selected in the range of 1 kHz to 1 MHz to measure ROS production in an aqueous solution containing Fe_3_O_4_ nanoparticles.

AEF with optimized frequency (1 Mhz) was administered to stimulate magnetite-bound Aβ fibril in vitro or in a transgenic AD mouse brain.

### 4.3. ROS Measurement

Six wells of solutions per group were used to measure ROS levels. The following solutions made with phosphate-buffered saline (PBS) were treated with 1 MHz, 1 V/cm AEF for 30 min: 1 µM APF; 80 µg/mL magnetite with 1 µM APF; 5 µM DHE; 80 µg/mL magnetite containing 5 µM DHE. The generation of ROS species was measured as fluorescence intensity using a plate reader set at the appropriate wavelength: APF, λex = 485 nm/λem = 510 nm; DHE, λex = 465 nm/λem = 590 nm.

To monitor the integrity of the Aβ-Fe_3_O_4_ fibril under an electric field, 20 µM thioflavin-T was used. Aβ solutions with or without magnetite, which were incubated over 144 h, were applied with a 1 MHz, 1 V/cm AEF in the brain phantom every 5 min over a period of 0 to 60 min. Four wells of each treated solution per group were analyzed for fluorescence intensity using a plate reader set at the appropriate wavelength ThT, λex = 430 nm/λem = 510 nm.

### 4.4. TEM Imaging

A 10 μL aliquot of either untreated or AEF-treated Aβ/magnetite or Aβ-only solutions was pipetted onto carbon/formvar-coated copper transmission electron microscopy (TEM) grids (01800, TED PELLA Inc., Redding, CA, USA), and excess liquid was removed to allow for the deposition of Aβ/magnetite aggregates. Each sample was observed using bio-TEM (HT 7700, Hitachi, Tokyo, Japan). Images were acquired at 100 kV acceleration voltage and 7.0 μA emission current.

### 4.5. Animal Model and AEF Treatment

All animal experiments were approved by the Institutional Animal Care and Use Committee (DCIA FCR-220928-21Y). All applicable international and institutional guidelines for the care and use of animals were followed. The 5xFAD [B6SJL-Tg(APPSwFl Lon, PSEN1∗M146L∗L286 V)6799Vas/Mmjax] and 3xTG [B6;129-Tg(APPSwe,tauP301L)1Lfa Psen1 tm1Mpm/Mmjax] transgenic AD mice were purchased from the Jackson Laboratory (Bar Harbor, ME, USA). A pilot animal experiment was performed on either 5xFAD (n = 5, aged 4 months) or 3xTG (n = 3, aged 6 months) mice. Non-Tg WT mice (n = 3, aged 4 months) were used as the control group.

A cocktail of 100 mg/kg of ketamine and 20 mg/kg of xylazine was injected intraperitoneally in mice to induce anesthesia. AEF was applied to the entire brain (PS) by securing each electrode over the haploid brain (Figure 8B). The measured voltage of AEF between the paired electrodes was either 0.7 or 1.4 V/cm at 1 MHz.

### 4.6. Sample Preparation and Histological Staining of Plaques

The samples used for histological staining were prepared as previously described [20]. Briefly, the animals were sacrificed and perfused intracardially with normal saline and then with approximately ∼50 mL of fixative containing 4% (wt/vol) paraformaldehyde in PBS (pH 7.4). The brains were harvested, postfixed in fresh fixative overnight, and subsequently placed in PBS. The samples used for histological staining were embedded in paraffin, sectioned at 10 μm thickness, and mounted onto glass slides. Mounted sections were stained with acidified 1% (wt/vol) potassium ferricyanide for Turnbull staining of Fe^2+^ iron, or with 0.2% Congo red in 50% alcohol for the staining of amyloid plaques using a modified Highman’s Congo red staining protocol [36]. Sections were stained with Congo red for approximately 10 min, and then the tissues were washed to remove unbound Congo red; subsequently, they were stained with hematoxylin for 2 min. The stained sections were examined under a microscope. The number of stained plaques in each section was counted manually. To determine the load of Aβ-plaque or iron, data from 4–6 mouse-brain sections, including the frontal cortex and hippocampus, were obtained from the early onset or aged mice, respectively, at ×200 magnification and used for statistical analysis.

### 4.7. Behavioral Tests

#### 4.7.1. Radial Arm Maze (RAM) Test

Non-Tg WT and AD mice underwent a RAM test to assess neurocognition before and after AEF treatment, as previously described [21]. Briefly, the maze consisted of a central chamber with eight equally spaced arms extending outward. One reward cup was placed on a platform at the distal end of each arm. Each mouse first underwent 10 min of habituation training trials in the RAM for 3 consecutive days. After acclimation, the mice were provided free access to all open arms for the duration of the testing session (two arms separated by 135° were baited), where the mice had to find the baited arms. The trial was terminated when the animal located the baited arms and consumed the food as a reward. The number of arms visited by the mice before visiting the two baited arms, including revisits, was counted. The result was considered correct, only if the mouse approached and ate the baited food cup. A visiting error was considered a spatial working memory error that occurred when a mouse re-entered an arm that was unbaited or had been previously baited. These performance measures were acquired before and after PS treatment, and the results were analyzed by analysis of variance (ANOVA).

#### 4.7.2. Novel Object Recognition (NOR) Test

The NOR test [37] was performed three days prior to AEF and at one month after treatment. Each mouse was presented with two similar objects during the first session, and then one of the two objects was replaced with a new object during the second session. The amount of time taken to explore the new object provided an index of memory recognition by calculating the difference in the measured time as a score. This tested the ability of the mouse to distinguish between objects. Typically, a normal mouse pays more attention to the novel object.

### 4.8. Safety Evaluation of AEF Treatment with Large-Volume Imaging by Tissue Clearing

Thy1-YFP-H transgenic mice purchased from the Jackson Lab (JAX#003782) were used to evaluate the safety of AEF treatment on brain tissues. All animal experiments were conducted in accordance with the guidelines for animal care and use as approved by the Institutional Animal Care and Use Committee of KIT (Study No. RS19003). Untreated and AEF-treated mice were anesthetized by an intraperitoneal injection of a saline solution containing Zoletil (Virbac, Caross, France) and Rompun (Bayer, Seoul, Republic of Korea). Brains were dissected and perfused mice with PBS and 4% paraformaldehyde. Whole brains were postfixed with 4% paraformaldehyde for 8 h at 4 °C. 2 mm thick coronal sections were cleared with Binaree Tissue Clearing Solution (Cat #HRTC-002, Daegu, Republic of Korea) at 37 °C for 3 days. Fluorescent images were obtained using Lightsheet Z1 (Carl Zeiss, Jena, Germany) from large-volume brain samples with a thickness interval of 6–7 μm. To visualize 3D images, multiple images were stacked and rendered using IMARIS (Bitplane AG, Schlieren, Switzerland).

### 4.9. Statistics

Data analysis was performed using the GraphPad Prism 6 software. Statistical analyses were performed using one-way ANOVA. Data are presented as means ± SD. The *p*-values of each group in the figure represent the confidence to estimate the mean of three experimental replicates ± SD. Student’s *t*-test was used to compare differences between the two groups. Differences were considered statistically significant at *p* < 0.05.

## 5. Conclusions

Herein, we report a transcranial electric field stimulation that effectively degrades iron deposits together with protein aggregates in 5xFAD and 3xTG AD mouse models using electric field-sensitized endogenous magnetite. Our results suggest that AEF stimulates deep brain tissue in the AD mouse brain, thereby facilitating memory improvement. Appropriate AEF offers a novel molecular targeting electroceutical treatment for AD.

## 6. Patents

A patent was applied on 21 September 2022.

## Figures and Tables

**Figure 1 ijms-24-09552-f001:**
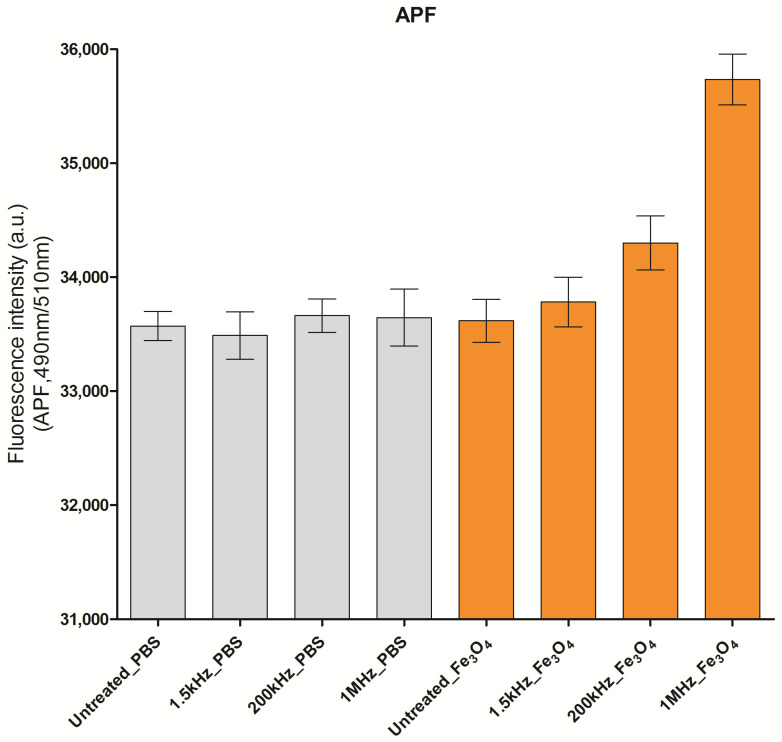
APF oxidant-based fluorescence measurement of hydroxyl radical, showing frequency dispersion of electric field in the generation of hydroxyl radicals from AEF-stimulated magnetite (Fe_3_O_4_) and PBS solutions with various frequencies of electric fields.

**Figure 2 ijms-24-09552-f002:**
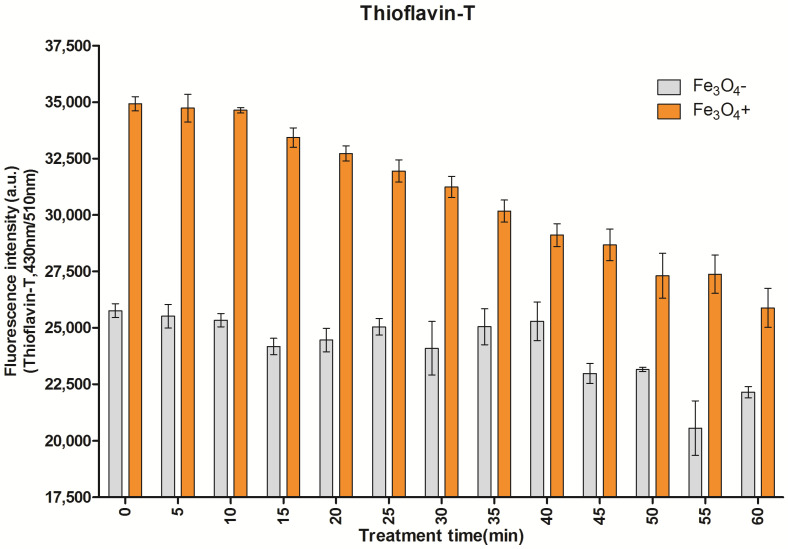
Graph of the measurements of thioflavin-T fluorescence (Tf-T) in both Aβ-magnetite and Aβ-alone fibrils following AEF treatment as a function of time. Observation of differential effects on fibril integration beginning 30 min following AEF treatment, at which the disintegration of the Aβ-magnetite fibril is apparent but Aβ-alone fibrils remain intact.

**Figure 3 ijms-24-09552-f003:**
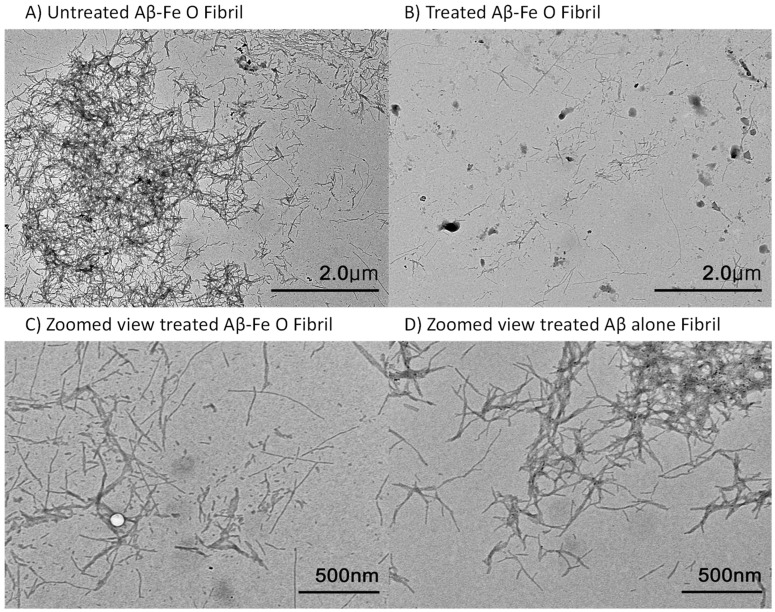
TEM imaging of Aβ-magnetite fibril (**A**,**C**) following AEF treatment showing the fragmentation of the fibril at 30 min following AEF treatment, compared to untreated control (**B**); intact state of Aβ-alone fibril (**D**). In (**C**), a bubble-like white artifact appeared due to a part that was not fully dried.

**Figure 4 ijms-24-09552-f004:**
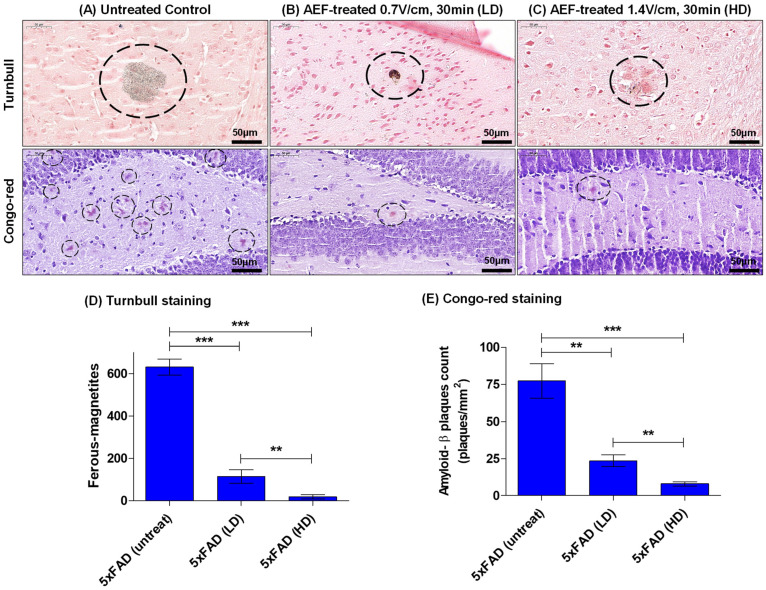
AEF treatment conducted on 5xFAD Alzheimer’s disease mouse model showing the breakup of insoluble Aβ-magnetite plaque in Turnbull staining (upper panel in (**A**–**C**)) and Congo red (lower panel in (**A**–**C**)) assays with corresponding quantitative measurements of ferrous iron oxide nanoparticles, magnetite (**D**), and Aβ plaque (**E**), respectively. Turnbull and Congo red staining were performed on tissue slides for each group (n = 3). ***: *p* < 0.001, **: *p* < 0.01.

**Figure 5 ijms-24-09552-f005:**
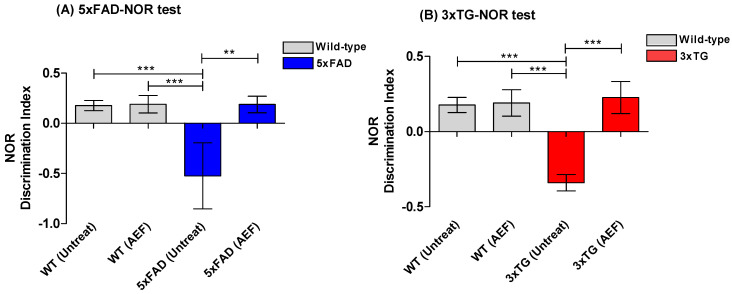
Results of novel object recognition (NOR) test for AD-mice (5xFAD) (**A**) and (3xTG) (**B**) species before and after AEF treatment. AD mice prior to AEF treatment (n = 8, pre-EFT) show a significant (*p* < 0.001) differential index (DI) compared to normal, wildtype (WT) mice; AEF-treated-mice group (5xFAD, n = 5; 3xTG, n = 3) demonstrates memory improvement compared to untreated pre-EFT AD mice. ***: *p* < 0.001, **: *p* < 0.01.

**Figure 6 ijms-24-09552-f006:**
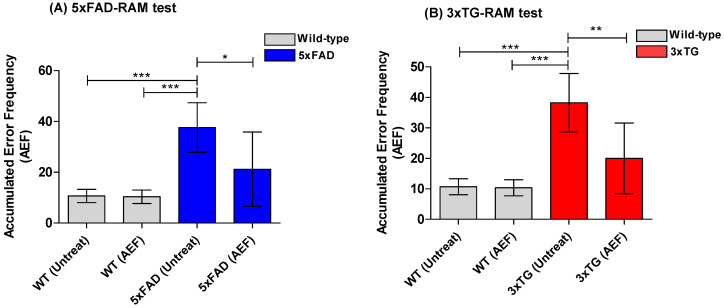
Results of radial arm maze (RAM) test conducted on AD-mice (5xFAD) (**A**) and (3xTG) (**B**) species before and after AEF treatment. AD mice prior to AEF treatment (n = 8, pre-EFT) show enhanced (*p* < 0.001) error frequencies compared to normal, wildtype (WT) mice; AEF-treated-mice group (5xFAD, n = 5; 3xTG, n = 3) demonstrates memory improvement (*p* < 0.001), compared to untreated pre-EFT AD mice. ***: *p* < 0.001, **: *p* < 0.01, *: *p* < 0.05.

**Figure 7 ijms-24-09552-f007:**
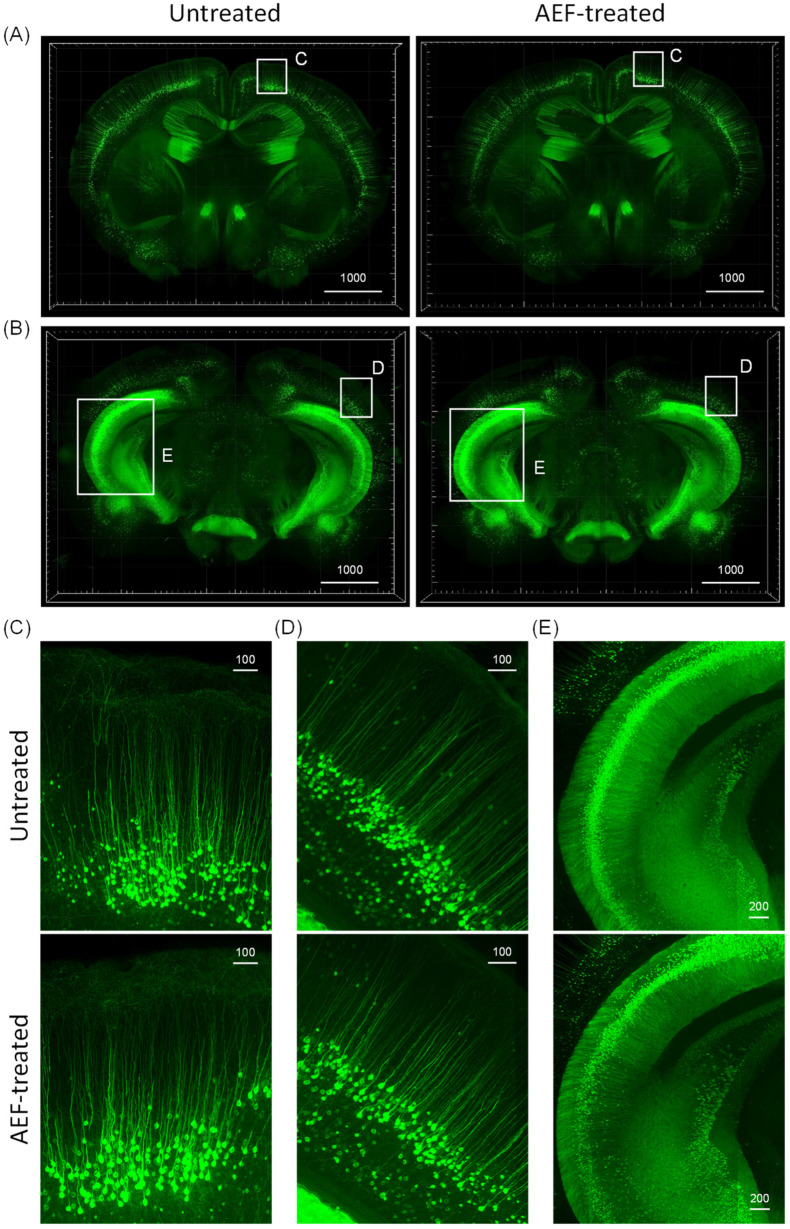
Comparison of AEF-treated and -untreated Thy1(CD90)-YFP (yellow fluorescent protein) transgenic mouse brain. Fluorescence images of the endogenous YFP signals from untreated and AEF-treated brains were obtained by large-volume imaging with a Lightsheet microscope using the tissue-clearing technique. Anterior (500 μm, (**A**)) and posterior (800 μm, (**B**)) parts of the coronal view of the brain are also visualized with maximum intensity projection that presented no topographical change in the neuronal structure after AEF treatment compared with untreated one. The distribution and morphology of Thy1-YFP-positive neurons can be observed in the zoomed-in images of the visual cortex (**C**), motor cortex (**D**), amygdala, and dentate gyrus (**E**) as highlighted with white box in A and B. Scale bars = 1000 μm, 200 μm, 100 μm.

**Figure 8 ijms-24-09552-f008:**
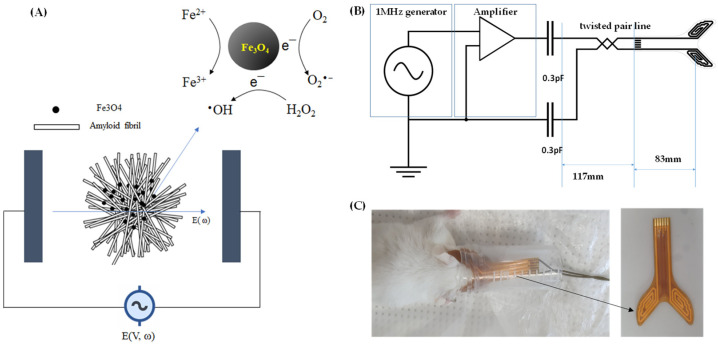
Schematic diagram of treatment with an alternating electric field (AEF) to generate an electro-Fenton reaction from a magnetite solution (**A**). A circuit diagram of generating AEF (**B**) and treatment on an Alzheimer’s disease mouse model with a capacitive-type electrode (**C**). The area of each anode or cathode electrode covers the entire haploid of the brain geometrically (**C**), resulting in the exposure of the electric field over the whole brain.

## Data Availability

All the data were presented in this article.

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
