# Peer review of "Capacitive Electrode-Based Electric Field Treatments on Redox-Toxic Iron Deposits in Transgenic AD Mouse Models: The Electroceutical Targeting of Alzheimer’s Disease Feasibility Study"

_ijms, 2023, doi:10.3390/ijms24119552_

Round 1

Reviewer 1 Report

Reviewer comments and suggestions

The authors in this study investigated electro-Fenton effects sensitized by an alternating electric field (AEF) on magnetite in an aqueous solution and a magnetite-bound amyloid fibril or in vivo protein deposition containing magnetite as a pilot treatment on transgenic AD mouse model. 

The results of the behavioral tests show an improvement in impaired cognitive function following AEF treatment on the AD mouse model. Finally, the authors concluded that, intraplaque magnetite exhibited degradation on the magnetite-bound amyloid fibril by electric-field sensitized electron release and ROS formation from magnetite and revealed a therapeutic effect on the pathological burden of AD and memory improvement in mice model

Overall, the manuscript was well written. However, a few concerns/comments needed to be explained/modified. 

  1. Line 18-21, Need to rewrite the first part of the abstract
  2. Line 39 which stage the author is discussing need to add point
  3. Line 59 the authors need to compressively discuss the cited references
  4. Line 70 please explore the study, only citing was not enough to understand
  5. Line 85 what was the point of discussing in relation to AD
  6. Comments for Figure 7 I could not find any differentiation between a and b in both group, please explain in a better way
  7. Comments for the discussion, Please add the first para consisting of what new findings from this study.
  8. Line 251-253 what was the possible reason or mechanism related to this therapy
  9. For the conclusion section Need to add more in the conclusion para, as it was not sufficient to highlight the importance for the common reader of your manuscript.
  10. Please check the references, the format still needs to be checked again.

Author Response

Dear reviewer,

We responded to all the question and comments with appropriate correction and rewriting All these amendments were displayed in blue (rewrote) or red (correction by English editing) color in revised manuscript.

Many thanks for this opportunity.

Jong-Ki Kim

Professor in Radiology and Biomedical Engineering

Daegu Catholic University, School of Medicine

Reviewer comments and suggestions

Reviewer 1

The authors in this study investigated electro-Fenton effects sensitized by an alternating electric field (AEF) on magnetite in an aqueous solution and a magnetite-bound amyloid fibril or in vivo protein deposition containing magnetite as a pilot treatment on transgenic AD mouse model.

The results of the behavioral tests show an improvement in impaired cognitive function following AEF treatment on the AD mouse model. Finally, the authors concluded that, intraplaque magnetite exhibited degradation on the magnetite-bound amyloid fibril by electric-field sensitized electron release and ROS formation from magnetite and revealed a therapeutic effect on the pathological burden of AD and memory improvement in mice model

Overall, the manuscript was well written. However, a few concerns/comments needed to be explained/modified.

Line 18-21, Need to rewrite the first part of the abstract

We rewrote first part of two paragraphs.

  • A capacitive electrode-based alternating electric field (AEF) was applied to a suspension of magnetite (Fe3O4) to measure field-sensitized reactive oxygen species (ROS) generation. The increase in ROS generation compared to the untreated control was both exposure-time and AEF-frequency dependent.

Line 39 which stage the author is discussing need to add point

We added stage number as follows in revised manuscript.

  • As the Braak stage increases, the tau pathology seems to spread to the forebrain, in-cluding the entorhinal cortex [2-3] (stage 3-4), and eventually results in the accumula-tion of toxic tauopathies in the nucleus basalis of Meynert (NBM), leading to clinically apparent cholinergic failure [4] (stage 5-6).

Line 59 the authors need to compressively discuss the cited references

We add explanatory note in revised manuscript,

  • Therefore, iron overload drives a series of events, including microglial activation by uptake of iron, inducing reactive astrocytes by ROS generation, promotion of Aβ plaque and tau tangles by iron binding, and ferroptosis-mediated neuronal or astrocyte loss aggravating the disease [16-19].

Line 70 please explore the study, only citing was not enough to understand,

We added relevant information for the crucial finding of previous work in revised manuscript,

  • Our prior proof-of-concept study targeting intraplaque magnetite demonstrated the curing effects on pathologies and cognitive decline via electron-release activity occur-ring from proton-stimulated magnetite [21]. Proton sensitized electron emission made breaking up covalent bonding in both Fe3O4-Aβ and protein matrix disrupting insoluble Aβ plaque that led to removal of toxic iron deposit and protein aggregates and induced cognitive improvements.

Line 85 what was the point of discussing in relation to AD

We rewrote this statement manifesting with discussing points in revised one,

  • Moreover, generation of free radicals could be accelerated in the magnetite solution under alternating magnetic field induction [28].

Comments for Figure 7 I could not find any differentiation between a and b in both group, please explain in a better way.

We rewrote Figure legend with a note telling no topographical change after AEF-treatment compared with untreated one.

  • Comparison of AEF-treated and -untreated Thy1(CD90)-YFP (yellow fluorescent protein) transgenic mouse brain. Fluorescence images of the endogenous YFP signals from untreated and AEF-treated brains were obtained by large-volume imaging with a Lightsheet microscope using the tissue-clearing technique. Anterior (500 μm, A) and posterior (800 μm, B) parts of the coronal view of the brain are also visualized with maximum intensity projection that presented no topographical change in neuronal structure after AEF-treatment compared with untreated one. The distribution and morphology of Thy1-YFP-positive neurons can be observed in the zoomed-in images of the visual cortex (C), motor cortex (D), amygdala and dentate gyrus (E). Scale bars = 1,000 μm, 200 μm, 100 μm.

Comments for the discussion, Please add the first para consisting of what new findings from this study.

We added a relevant statement about this.

  • In this study we found that frequency-specific alternating electric field was able to sensitize electron emissions from the magnetite surface, inducing electro-Fenton effect in aqueous solution, to disrupt insoluble amyloid plaque containing redox-toxic magnetite.

Line 251-253 what was the possible reason or mechanism related to this therapy,

Regarding proton stimulation therapy (PS), we found that PS-derived removal of redox-toxic iron deposit together with protein aggregates triggered a cascade of disease modifying effects including control of ferroptosis, neuroinflammation, tau spreading and reactive astrocytes, as well as restoration of neurogenesis eventually helpful to cognitive improvement. These data have been prepared to write another manuscript for publication in other journal. We try to add some relevant statement following original Line 251-253 in revised manuscript.

  • This result is consistent with the results of proton-stimulation (PS) therapy targeting intraplaque magnetite, as described in our previous study [15, 21], which may elicit similar biological end-effects from both brain-stimulation tools targeting common intraplaque magnetite. Ongoing PS study revealed that a cascade of disease modifying events, such as down-regulating neuroinflammation, reactive astrocyte and blocking tau spreading helpful to cognitive improvement, were triggered following PS-derived removal of toxic iron deposit together with amyloid plaque (Data were prepared for publication in another Journal).

For the conclusion section Need to add more in the conclusion para, as it was not sufficient to highlight the importance for the common reader of your manuscript.

We added more statements highlighting the importance of this study with correction of some typo.

  • Herein, we report a transcranial electric field stimulation that effectively degrades iron deposits together with protein aggregates in 5xFAD and 3xTG AD mouse models using electric field-sensitized endogenous magnetite. Our results suggest that AEF stimulates deep brain tissue in the AD mouse brain, thereby facilitating memory improvement. Appropriate AEF offers a novel molecular targeting electroceutical treatment for AD.

Please check the references, the format still needs to be checked again.

We corrected a number of space error in author name or unnecessary (,) after Journal name.

Reviewer 2 Report

The manuscript entitled “Capacitive-electrode based Electric field treatments on redox-toxic iron deposits in transgenic AD mouse models: The electroceutical targeting of Alzheimer’s disease feasibility study” by Choi and co-authors reported an increase in ROS production in response to alternating electric field. Authors have compared the ROS generation with control. The behavioral tests results revealed an improvement in impaired cognitive function following AEF treatment on the AD mouse model. Further, tissue clearing and 3D-imaging results indicated no induced damage to neuronal structures of brain tissue. The main take away of manuscript is that AEF study targeting magnetite illustrated the degradation on the magnetite-bound amyloid fibril by electric-field sensitized electron release and ROS formation. Overall, the manuscript is written well and work is performed systematically. Although, I have a few queries that need to be addressed prior to consideration of manuscript for publication. I recommend a minor revision of the manuscript and here are my suggestions for the manuscript:-

1.      Authors need to reframe the figures like in Figure 1, labeling are on the edge of figure and font types are also different than other figure labelings.

2.      Authors need to provide the clear outcomes and rational of study in abstract.

3.      Authors should discuss the outcomes of latest studies in discussion.

4.      The manuscript is written well. Although, there are several grammatical, formatting and typos errors. So, authors should thoroughly revise the manuscript to fix the typos and grammatical mistakes. 

The manuscript entitled “Capacitive-electrode based Electric field treatments on redox-toxic iron deposits in transgenic AD mouse models: The electroceutical targeting of Alzheimer’s disease feasibility study” by Choi and co-authors reported an increase in ROS production in response to alternating electric field. Authors have compared the ROS generation with control. The behavioral tests results revealed an improvement in impaired cognitive function following AEF treatment on the AD mouse model. Further, tissue clearing and 3D-imaging results indicated no induced damage to neuronal structures of brain tissue. The main take away of manuscript is that AEF study targeting magnetite illustrated the degradation on the magnetite-bound amyloid fibril by electric-field sensitized electron release and ROS formation. Overall, the manuscript is written well and work is performed systematically. Although, I have a few queries that need to be addressed prior to consideration of manuscript for publication. I recommend a minor revision of the manuscript and here are my suggestions for the manuscript:-

1.      Authors need to reframe the figures like in Figure 1, labeling are on the edge of figure and font types are also different than other figure labelings.

2.      Authors need to provide the clear outcomes and rational of study in abstract.

3.      Authors should discuss the outcomes of latest studies in discussion.

4.      The manuscript is written well. Although, there are several grammatical, formatting and typos errors. So, authors should thoroughly revise the manuscript to fix the typos and grammatical mistakes. 

Author Response

Dear reviewer,

We responded to all the question and comments with appropriate correction and rewriting All these amendments were displayed in blue (rewrote) or red (correction by English editing) color in revised manuscript.

Many thanks for this opportunity.

Jong-Ki Kim

Professor in Radiology and Biomedical Engineering

Daegu Catholic University, School of Medicine

Reviewer comments and suggestions

The manuscript entitled “Capacitive-electrode based Electric field treatments on redox-toxic iron deposits in transgenic AD mouse models: The electroceutical targeting of Alzheimer’s disease feasibility study” by Choi and co-authors reported an increase in ROS production in response to alternating electric field. Authors have compared the ROS generation with control. The behavioral tests results revealed an improvement in impaired cognitive function following AEF treatment on the AD mouse model. Further, tissue clearing and 3D-imaging results indicated no induced damage to neuronal structures of brain tissue. The main take away of manuscript is that AEF study targeting magnetite illustrated the degradation on the magnetite-bound amyloid fibril by electric-field sensitized electron release and ROS formation. Overall, the manuscript is written well and work is performed systematically. Although, I have a few queries that need to be addressed prior to consideration of manuscript for publication. I recommend a minor revision of the manuscript and here are my suggestions for the manuscript:-

  1. Authors need to reframe the figures like in Figure 1, labeling are on the edge of figure and font types are also different than other figure labelings.

We corrected labelling inside figure frame and made fonts same as other figures.

  1. Authors need to provide the clear outcomes and rational of study in abstract.

We rewrote abstract accordingly with outcomes and rationale of study.

  • Abstract: Iron accumulation in the brain accelerates Alzheimer’s disease progression. To cure iron toxicity, we assessed the therapeutic effects of non-contact transcranial electric field stimulation to the brain on toxic iron deposit in either Aβ fibril structure or Aβ plaque in a mouse model of Alzheimer’s disease (AD) as a pilot study. A capacitive electrode-based alternating electric field (AEF) was applied to a suspension of magnetite (Fe3O4) to measure field-sensitized reactive oxygen species (ROS) generation. The increase in ROS generation compared to the untreated control was both exposure-time and AEF-frequency dependent. The frequency-specific exposure of AEF to 0.7–1.4 V/cm on a magnetite-bound Aβ-fibril or a transgenic Alzheimer’s disease (AD) mouse model revealed the degradation of the Aβ fibril or the removal of the Aβ-plaque burden and ferrous magnetite compared to the untreated control. The results of the behavioral tests show an improvement in impaired cognitive function following AEF treatment on the AD mouse model. Tissue clearing and 3D-imaging analysis revealed no induced damage to the neuronal structures of normal brain tissue following AEF treatment. In conclusion, our results suggest that the effective degradation of magnetite-bound amyloid fibrils or plaques in the AD brain by electro-Fenton effect from electric field-sensitized magnetite offers a potential electroceutical treatment option for AD.

  1. Authors should discuss the outcomes of latest studies in discussion.

 We rewrote discussion section accordingly, and additional in conclusion section.

Beginning with founding of this study and adding mechanism of AEF therapy in comparison with our previous Proton stimulation therapy study,

  • Discussion

In this study we found that frequency-specific alternating electric field was able to sensitize electron emissions from the magnetite surface, inducing electro-Fenton effect in aqueous solution, to disrupt insoluble amyloid plaque containing redox-toxic magnetite. The exact origin of the magnetite integration into the Aβ plaque remains unknown. A few studies have suggested that these are generated by interaction between Aβ and ferritin, which contains iron oxide nanoparticles [29, 30]. However, not all Aβ-plaques contain magnetite. The results from our previous study [21] estimated that > 70% of Aβ plaques contain magnetite, which is concurrent to other studies [31]. AEF-sensitized magnetite produced high levels of hydroxyl radicals, particularly at a higher frequency of the applied electric field. This effect might be due to the electro-Fenton effect on the magnetite solution caused by the static electric field [26], suggesting that the potential induction of the bipolar electrochemistry [26] is transiently set up by the rapidly changing field and dielectric dispersion property of magnetite [32]. In turn, the enhancement of the electro-Fenton effect can be ascribed to the increased electron transport activity and AC conductivity of magnetite with the increasing AEF frequency [33-34]. AEF induces electron release from the magnetite surface and formation of ROS near the magnetite surface within the context of insoluble Aβ plaque. Therefore magnetite-derived electrons and hydroxyl radicals are prone to be consumed mainly in breaking Aβ plaque-magnetite bonds (Figures 3,4), because the Aβ plaque matrix is covalently bonded with the magnetite.

Because AEF can be delivered to the deep brain in a non-contact transcranial manner with capacitive electrodes, targeting magnetite particles in protein aggregates using AEF therapy might serve as an electroceutical treatment option for AD patients. Hence, testing the AEF-driven electro-Fenton effect on the magnetite fibril in vitro or within AD deep brain in vivo is crucial. The differential time values for inducing a disruption between Aβ-magnetite and Aβ-alone fibrils, as shown in the Tf-T fluorescence data presented in Figure 3, were used to determine 30 min as the optimum exposure time for AEF treatment while showing the electro-Fenton effects of magnetite on the fibril. The fragmentation and disruption of the fibrils, as shown in the TEM images in Figure 4, suggests a direct effect of AEF-driven electron emissions from the magnetite surface on the covalent bonding of Aβ proteins, as observed in our previous studies [15, 21]. Thus, the generation of both electron and hydroxyl radicals from the AEF-exposed magnetite may not only break the bond of the Aβ–magnetite complex, but also the Aβ protein matrix in the fibril. Based on these results, we administered AEF to a transgenic AD mouse model at different doses: 0.7 and 1.4 V/cm for 30 min. The dose-dependent degradation and reduction in both intraplaque iron species and Aβ plaque, as shown in Figure 5. The entorhinal–hippocampal section demonstrated the effects caused by the interaction of AEF with intraplaque magnetite in the AEF-treated mouse model. This result indicates the potential therapeutic effects of transcranial AEF administration in the deep brain, potentially such as the locus coeruleus-norepinephrine system, which is an important target site for direct electroceutical stimulation in early onset AD [20]. This result is consistent with the results of proton-stimulation (PS) therapy that targeted intraplaque magnetite, as described in our previous study [15, 21]. These studies together suggest that similar biological end-effects are elicited by both brain-stimulation tools targeting the common intraplaque magnetite. Moreover, the AEF-driven removal of toxic iron deposits suggests the potential beneficial effects of AEF-treatment in preventing either monoamine oxidase B activity or reactive astrocytes formation, which has been implicated as major culprit to cause AD [35]. Ongoing PS study revealed that a cascade of disease modifying events, such as down-regulating neuroinflammation, reactive astrocyte and blocking tau spreading helpful to cognitive improvement, were triggered following PS-derived removal of toxic iron deposit together with amyloid plaque (Data were prepared for publication in another Journal). AD mice demonstrated clearly impaired memory function compared to WT control mice (Figure 6 and -7), and both AD-mouse species demonstrated an improvement in cognitive function following AEF treatment (p < 0.001). Delivery of different types of physical energy on magnetite involved common electron emissions from the surface of iron oxide nanoparticles, which induced the fragmentation of insoluble Aβ plaque containing ferrous magnetite. The degradation of Aβ plaque into smaller-sized peptides and isolated magnetite increase the solubility of plaque and facilitated microglial phagocytosis-mediated clearance.

WT-normal mice did not show significant differences in behavioral task before and after AEF treatment, suggesting that the therapy did not induce any detrimental neuronal damage, which was consistent with the results of the tissue-clearing based 3D-imaging analysis of AEF-treated brain tissue.

  • Conclusions

Herein, we report a transcranial electric field stimulation that effectively degrades iron deposits together with protein aggregates in 5xFAD and 3xTG AD mouse models using electric field-sensitized endogenous magnetite. Our results suggest that AEF stimulates deep brain tissue in the AD mouse brain, thereby facilitating memory improvement. Appropriate AEF offers a novel molecular targeting electroceutical treatment for AD.

  1. The manuscript is written well. Although, there are several grammatical, formatting and typos errors. So, authors should thoroughly revise the manuscript to fix the typos and grammatical mistakes. 

We corrected grammatical, formatting and typos by additional English editing service, all the amendments are displayed in red color in revised manuscript.

Typically as shown in response to comment 3
